# Learning to Deceive Knowledge Graph Augmented Models via Targeted Perturbation

**Mrigank Raman**[1*], **Aaron Chan**[2†], **Siddhant Agarwal**[3*†], **Peifeng Wang**[2], **Hansen Wang**[4*],
**Sungchul Kim**[5], **Ryan Rossi**[5], **Handong Zhao**[5], **Nedim Lipka**[5], **Xiang Ren**[2]
[1]Indian Institute of Technology, Delhi, [2]University of Southern California,
[3] Indian Institute of Technology, Kharagpur, [4]Tsinghua University, [5]Adobe Research
`mt1170736@iitd.ac.in`, `{chanaaro,peifengw,xiangren}@usc.edu`,
`agarwalsiddhant10@iitkgp.ac.in`, `wang-hs17@mails.tsinghua.edu.cn`,
`{sukim,ryrossi,hazhao,lipka}@adobe.com`

## Abstract

Knowledge graphs (KGs) have helped neural models improve performance on various knowledge-intensive tasks, like question answering and item recommendation. By using attention over the KG, such KG-augmented models can also "explain" which KG information was most relevant for making a given prediction. In this paper, we question whether these models are really behaving as we expect. We show that, through a reinforcement learning policy (or even simple heuristics), one can produce deceptively perturbed KGs, which maintain the downstream performance of the original KG while significantly deviating from the original KG's semantics and structure. Our findings raise doubts about KG-augmented models' ability to reason about KG information and give sensible explanations.

## 1 Introduction

Recently, neural reasoning over knowledge graphs (KGs) has emerged as a popular paradigm in machine learning and natural language processing (NLP). *KG-augmented models* have improved performance on a number of knowledge-intensive downstream tasks: for question answering (QA), the KG provides context about how a given answer choice is related to the question (Lin et al., 2019; Feng et al., 2020; Lv et al., 2020; Talmor et al., 2018); for item recommendation, the KG mitigates data sparsity and cold start issues (Wang et al., 2018b; 2019a;b; 2018a). Furthermore, by using attention over the KG, such models aim to explain which KG information was most relevant for making a given prediction (Lin et al., 2019; Feng et al., 2020; Wang et al., 2018b; 2019b; Cao et al., 2019; Gao et al., 2019). Nonetheless, the process in which KG-augmented models reason about KG information is still not well understood. It is assumed that, like humans, KG-augmented models base their predictions on meaningful KG paths and that this process is responsible for their performance gains (Lin et al., 2019; Feng et al., 2020; Gao et al., 2019; Song et al., 2019).

In this paper, we question if existing KG-augmented models actually use KGs in this human-like manner. We study this question primarily by measuring model performance when the KG's semantics and structure have been perturbed to hinder human comprehension. To perturb the KG, we propose four perturbation heuristics and a reinforcement learning (RL) based perturbation algorithm. Surprisingly, for KG-augmented models on both commonsense QA and item recommendation, we find that the KG can be extensively perturbed with little to no effect on performance. This raises doubts about KG-augmented models' use of KGs and the plausibility of their explanations.

## 2 Problem Setting

Our goal is to investigate whether KG-augmented models and humans use KGs similarly. Since KGs are human-labeled, we assume that they are generally accurate and meaningful to humans. Thus, across different perturbation methods, we measure model performance when every edge in the KG has been perturbed to make less sense to humans. To quantify the extent to which the KG has been perturbed, we also measure both semantic and structural similarity between the original

---

[*]Work done while MR, SA and HW interned remotely at USC. Code and data are available at `https://github.com/INK-USC/deceive-KG-models`.

[†]Equal contribution.

KG and perturbed KG. If original-perturbed KG similarity is low, then a human-like KG-augmented model should achieve worse performance with the perturbed KG than with the original KG. Furthermore, we evaluate the plausibility of KG-augmented models' explanations when using original and perturbed KGs, by asking humans to rate these explanations' readability and usability.

**Notation** Let $\mathcal{F}_\theta$ be an KG-augmented model, and let $(X_{\text{train}}, X_{\text{dev}}, X_{\text{test}})$ be a dataset for some downstream task. We denote a KG as $\mathcal{G} = (\mathcal{E}, \mathcal{R}, \mathcal{T})$, where $\mathcal{E}$ is the set of entities (nodes), $\mathcal{R}$ is the set of relation types, and $\mathcal{T} = \{(e_1, r, e_2) \mid e_1, e_2 \in \mathcal{E}, \; r \in \mathcal{R}\}$ is the set of facts (edges) composed from existing entities and relations (Zheng et al., 2018). Let $\mathcal{G}' = (\mathcal{E}, \mathcal{R}', \mathcal{T}')$ be the KG obtained after perturbing $\mathcal{G}$, where $\mathcal{R}' \subseteq \mathcal{R}$ and $\mathcal{T}' \neq \mathcal{T}$. Let $f(\mathcal{G}, \mathcal{G}')$ be a function that measures similarity between $\mathcal{G}$ and $\mathcal{G}'$. Let $g(\mathcal{G})$ be the downstream performance when evaluating $\mathcal{F}_\theta$ on $X_{\text{test}}$ and $\mathcal{G}$. Also, let $\oplus$ denote the concatenation operation, and let $\mathcal{N}_L(e)$ denote the set of $L$-hop neighbors for entity $e \in \mathcal{E}$.

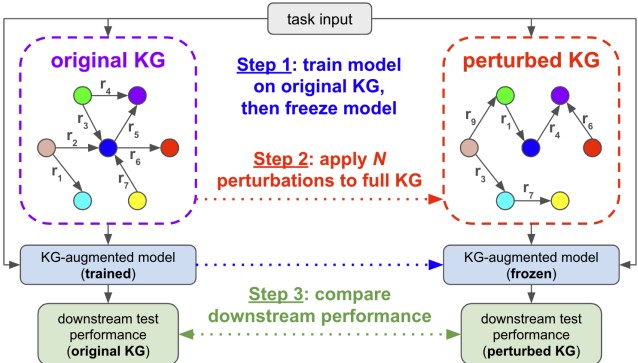

Figure 1: **Proposed KG Perturbation Framework.** Our procedure consists of three main steps: **(1)** train the KG-augmented model on the original KG, then freeze the model; **(2)** obtain the perturbed KG by applying $N = |\mathcal{T}|$ perturbations to the full original KG; and **(3)** compare the perturbed KG's downstream performance to that of the original KG.

**High-Level Procedure** First, we train $\mathcal{F}_\theta$ on $X_{\text{train}}$ and $\mathcal{G}$, then evaluate $\mathcal{F}_\theta$ on $X_{\text{test}}$ and $\mathcal{G}$ to get the original performance $g(\mathcal{G})$. Second, we freeze $\mathcal{F}_\theta$, then perturb $\mathcal{G}$ to obtain $\mathcal{G}'$. Third, we evaluate $\mathcal{F}_\theta$ on $X_{\text{test}}$ and $\mathcal{G}'$ to get the perturbed performance $g(\mathcal{G}')$. Finally, we measure $g(\mathcal{G}) - g(\mathcal{G}')$ and $f(\mathcal{G}, \mathcal{G}')$ to assess how human-like $\mathcal{F}_\theta$'s reasoning process is. This procedure is illustrated in Fig. 1. In this paper, we consider two downstream tasks: commonsense QA and item recommendation.

**Commonsense QA** Given a question $x$ and a set of $k$ possible answers $\mathcal{A} = \{y_1, ..., y_k\}$, the task is to predict a compatibility score for each $(x, y)$ pair, such that the highest score is predicted for the correct answer. In commonsense QA, the questions are designed to require commonsense knowledge which is typically unstated in natural language, but more likely to be found in KGs (Talmor et al., 2018). Let $\mathcal{F}_\phi^{\text{text}}$ be a text encoder (Devlin et al., 2018), $\mathcal{F}_\psi^{\text{graph}}$ be a graph encoder, and $\mathcal{F}_\xi^{\text{cls}}$ be an MLP classifier, where $\phi, \psi, \xi \subset \theta$. Let $\mathcal{G}_{(x,y)}$ denote a subgraph of $\mathcal{G}$ consisting of entities mentioned in text sequence $x \oplus y$, plus their corresponding edges. We start by computing a text embedding $\mathbf{h}_{\text{text}} = \mathcal{F}_\phi^{\text{text}}(x \oplus y)$ and a graph embedding $\mathbf{h}_{\text{graph}} = \mathcal{F}_\phi^{\text{graph}}(\mathcal{G}_{(x,y)})$. After that, we compute the score for $(x, y)$ as $S_{(x,y)} = \mathcal{F}_\xi^{\text{cls}}(\mathbf{h}_{\text{text}} \oplus \mathbf{h}_{\text{graph}})$. Finally, we select the highest scoring answer: $y_{\text{pred}} = \arg\max_{y \in A} S_{(x,y)}$. KG-augmented commonsense QA models vary primarily in their design of $\mathcal{F}_\psi^{\text{graph}}$. In particular, path-based models compute the graph embedding by using attention to selectively aggregate paths in the subgraph. The attention scores can help explain which paths the model focused on most for a given prediction (Lin et al., 2019; Feng et al., 2020; Santoro et al., 2017).

**Item Recommendation** We consider a set of users $\mathcal{U} = \{u_1, u_2, ..., u_m\}$, a set of items $\mathcal{V} = \{v_1, v_2, ..., v_n\}$, and a user-item interaction matrix $\mathbf{Y} \in \mathbb{R}^{m \times n}$ with entries $y_{uv}$. If user $u$ has been observed to engage with item $v$, then $y_{uv} = 1$; otherwise, $y_{uv} = 0$. Additionally, we consider a KG $\mathcal{G}$, in which $\mathcal{R}$ is the set of relation types in $\mathcal{G}$. In $\mathcal{G}$, nodes are items $v \in \mathcal{V}$, and edges are facts of the form $(v, r, v')$, where $r \in \mathcal{R}$ is a relation. For the zero entries in $\mathbf{Y}$ (i.e., $y_{uv} = 0$), our task is to predict a compatibility score for user-item pair $(u, v)$, indicating how likely user $u$ is to want to engage with item $v$. We represent each user $u$, item $v$, and relation $r$ as embeddings $\mathbf{u}, \mathbf{v}$, and $\mathbf{r}$, respectively. Given a user-item pair $(u, v)$, its compatibility score is computed as $\langle \mathbf{u}, \mathbf{v} \rangle$, the inner product between $\mathbf{u}$ and $\mathbf{v}$. KG-augmented recommender systems differ mainly in how they use $\mathcal{G}$ to compute $\mathbf{u}$ and $\mathbf{v}$. Generally, these models do so by using attention to selectively aggregate items/relations in $\mathcal{G}$. The attention scores can help explain which items/relations the model found most relevant for a given prediction (Wang et al., 2018b; 2019b).

## 3 KG SIMILARITY METRICS

To measure how much the perturbed KG has deviated from the original KG, we propose several metrics for capturing semantic (ATS) and structural (SC2D, SD2) similarity between KGs.

**Aggregated Triple Score (ATS)** ATS measures semantic similarity between two KGs. Let $s_{\mathcal{G}}$ be an edge (triple) scoring function, such that $s_{\mathcal{G}}(e_1, r, e_2)$ measures how likely edge $(e_1, r, e_2)$ is to exist in $\mathcal{G}$. Also, assume $s_{\mathcal{G}}$ has been pre-trained on $\mathcal{G}$ for link prediction. Then, ATS is defined as $f_{\text{ATS}}(\mathcal{G}, \mathcal{G}') = \frac{1}{|\mathcal{T}'|} \sum_{(e_1, r, e_2) \in \mathcal{T}'} s_{\mathcal{G}}(e_1, r, e_2) \in [0, 1]$, which denotes the mean $s_{\mathcal{G}}$ score across all edges in $\mathcal{G}'$. Intuitively, if a high percentage of edges in $\mathcal{G}'$ are also likely to exist in $\mathcal{G}$ (i.e., high ATS), then we say that $\mathcal{G}'$ and $\mathcal{G}$ have high semantic similarity. $s_{\mathcal{G}}$ is task-specific, as KGs from different tasks may differ greatly in semantics. For commonsense QA, we use the $s_{\mathcal{G}}$ from Li et al. (2016); for item recommendation, we use the $s_{\mathcal{G}}$ from Yang et al. (2015). While ATS captures semantic KG differences, it is not sensitive to KG connectivity structure. Note that $f_{\text{ATS}}(\mathcal{G}, \mathcal{G})$ may not equal 1, since $s_{\mathcal{G}}$ may not perfectly generalize to KGs beyond those it was trained on.

**Similarity in Clustering Coefficient Distribution (SC2D)** SC2D measures structural similarity between two KGs and is derived from the local clustering coefficient (Saramäki et al., 2007; Onnela et al., 2005; Fagiolo, 2007). For a given entity in $\mathcal{G}$ (treated here as undirected), the local clustering coefficient is the fraction of possible triangles through the entity that exist (i.e., how tightly the entity's neighbors cluster around it). For entity $e_i \in \mathcal{E}$, the local clustering coefficient is defined as $c_i = 2\text{Tri}(e_i)/(\deg(e_i)(\deg(e_i) - 1))$, where $\text{Tri}(e_i)$ is the number of triangles through $e_i$, and $\deg(e_i)$ is the degree of $e_i$. For each relation $r \in \mathcal{R}$, let $\mathcal{G}^r$ be the subgraph of $\mathcal{G}$ consisting of all edges in $\mathcal{T}$ with $r$. That is, $\mathcal{G}^r = (\mathcal{E}, r, \mathcal{T}')$, where $\mathcal{T}' = \{(e, r, e') \mid e, e' \in \mathcal{E}\}$. Let $\mathbf{c}^r$ denote the $|\mathcal{E}|$-dimensional clustering coefficient vector for $\mathcal{G}^r$, where the $i$th element of $\mathbf{c}^r$ is $c_i$. Then, the mean clustering coefficient vectors for $\mathcal{G}$ and $\mathcal{G}'$ are $\mathbf{c}_o = \frac{1}{|\mathcal{R}|} \sum_{r \in \mathcal{R}} \mathbf{c}^r$ and $\mathbf{c}_p = \frac{1}{|\mathcal{R}'|} \sum_{r \in \mathcal{R}'} \mathbf{c}^r$, respectively. SC2D is defined as $f_{\text{SC2D}}(\mathcal{G}, \mathcal{G}') = 1 - \frac{\|\mathbf{c}_o - \mathbf{c}_p\|_2}{\|\mathbf{c}_o - \mathbf{c}_p\|_2 + 1} \in [0, 1]$, with higher value indicating higher similarity.

**Similarity in Degree Distribution (SD2)** SD2 also measures structural similarity between two KGs, while addressing SC2D's ineffectiveness when the KGs' entities have tiny local clustering coefficients (e.g., the item KG used by recommender systems is roughly bipartite). In such cases, SC2D is always close to one regardless of perturbation method, thus rendering SC2D useless. Let $\mathbf{d}^r$ denote the $|\mathcal{E}|$-dimensional degree vector for $\mathcal{G}^r$, where the $i$th element of $\mathbf{d}^r$ is $\deg(e_i)$. Then, the mean degree vectors for $\mathcal{G}$ and $\mathcal{G}'$ are $\mathbf{d}_o = \frac{1}{|\mathcal{R}|} \sum_{r \in \mathcal{R}} \mathbf{d}^r$ and $\mathbf{d}_p = \frac{1}{|\mathcal{R}'|} \sum_{r \in \mathcal{R}'} \mathbf{d}^r$, respectively. SD2 is defined as $f_{\text{SD2}}(\mathcal{G}, \mathcal{G}') = 1 - \frac{\|\mathbf{d}_o - \mathbf{d}_p\|_2}{\|\mathbf{d}_o - \mathbf{d}_p\|_2 + 1} \in [0, 1]$, with higher value indicating higher similarity.

## 4 METHODS FOR TARGETED KG PERTURBATION

We aim to study how a KG's semantics and structure impact KG-augmented models' downstream performance. To do so, we measure model performance in response to various forms of targeted KG perturbation. While a KG's semantics can be perturbed via its relation types, its structure can be perturbed via its edge connections. Therefore, we design five methods — four heuristic, one RL — for perturbing KG relation types, edge connections, or both (Fig. 2).

### 4.1 HEURISTIC-BASED KG PERTURBATION

Our four KG perturbation heuristics are as follows: **Relation Swapping (RS)** randomly chooses two edges from $\mathcal{T}$ and swaps their relations. **Relation Replacement (RR)** randomly chooses an edge $(e_1, r_1, e_2) \in \mathcal{T}$, then replaces $r_1$ with another relation $r_2 = \arg\min_{r \in \mathcal{R}} s_{\mathcal{G}}(e_1, r, e_2)$. **Edge Rewiring (ER)** randomly chooses an edge $(e_1, r, e_2) \in \mathcal{T}$, then replaces $e_2$ with another entity $e_3 \in \mathcal{E} \setminus \mathcal{N}_1(e_1)$. **Edge Deletion (ED)** randomly chooses an edge $(e_1, r, e_2) \in \mathcal{T}$ and deletes it. For ED, perturbing all edges means deleting all but 10 edges.

### 4.2 RL-BASED KG PERTURBATION

We introduce an RL-based approach for perturbing the KG. Given a KG, $\mathcal{G}$, we train a policy to output a perturbed KG, $\mathcal{G}'$, such that ATS, $f_{\text{ATS}}(\mathcal{G}, \mathcal{G}')$, is minimized, while downstream performance, $g(\mathcal{G}')$, is maximized. Specifically, the RL agent is trained to perturb $\mathcal{G}$ via relation replacement, so we call our algorithm **RL-RR**. Because the agent is limited to applying $N = |\mathcal{T}|$ perturbations to $\mathcal{G}$, our RL problem is framed as a finite horizon Markov decision process. In the rest of this section, we define the actions, states, and reward in our RL problem, then explain how RL-RR is implemented.

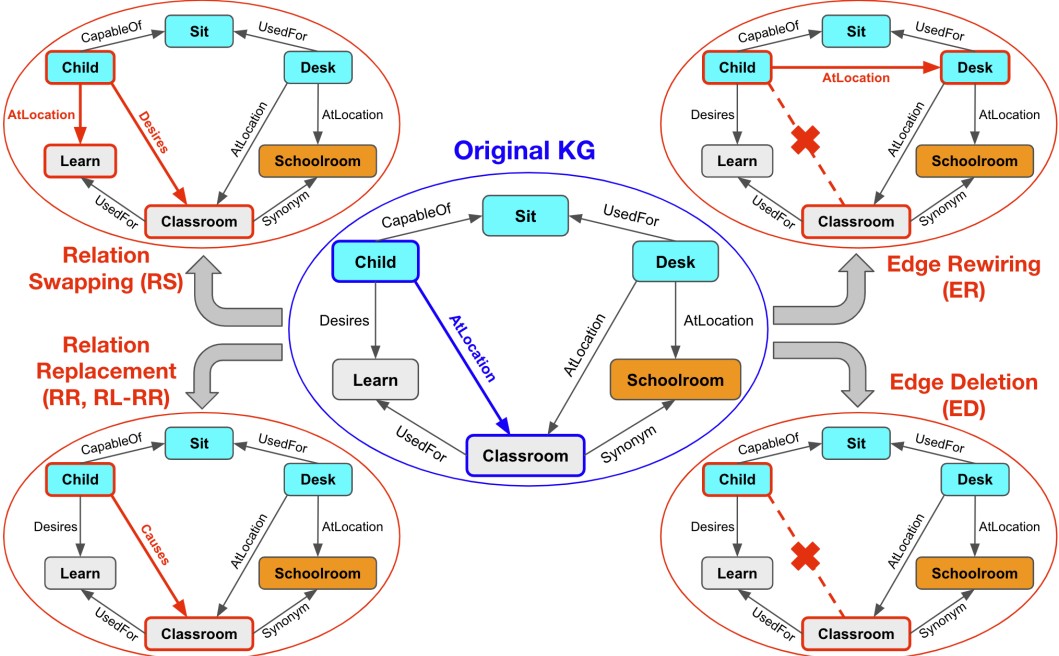

Figure 2: **Proposed KG Perturbation Methods.** We propose four heuristic-based perturbation methods and one RL-based perturbation method. In this diagram, we consider example edge (Child, AtLocation, Classroom) within a subgraph of the original ConceptNet KG (shown in blue). We illustrate how this edge (and possibly other edges) changes in response to different perturbation methods (shown in red). Unlike the heuristic-based methods (RS, RR, ER, ED), the RL-based method (RL-RR) is trained to maximize downstream performance and minimize original-perturbed KG semantic similarity.

**Actions** The action space consists of all possible relation replacements in $\mathcal{G}$, i.e., replacing $(e_1, r, e_2) \in \mathcal{T}$ with $(e_1, r', e_2)$. Since having such a large action space poses computational issues, we decouple each action into a sequence of three *subactions* and operate instead in this smaller subaction space. Hence, a perturbation action at time step $t$ would be $a_t = (a_t^{(0)}, a_t^{(1)}, a_t^{(2)})$. Namely, $a_t^{(0)}$ is sampling entity $e_1 \in \mathcal{E}$; $a_t^{(1)}$ is selecting edge $(e_1, r, e_2) \in \mathcal{T}$; and $a_t^{(2)}$ is selecting relation $r' \in \mathcal{R}$ to replace $r$ in $(e_1, r, e_2)$. To make the policy choose low-ATS perturbations, we further restrict the $a_t^{(2)}$ subaction space to be the $K$ subactions resulting in the lowest ATS. Note that each $a_t^{(i)}$ is represented by its corresponding pre-trained TransE (Bordes et al., 2013) entity, relation, or edge embedding in $\mathcal{G}$. Since these TransE embeddings are not updated by the perturbation policy, we use $a_t^{(i)}$ to refer to both the subaction and subaction embedding. Meanwhile, $a_t$ does not have any representation besides its constituent subaction embeddings.

**States** The state space is the set of all $\mathcal{G}'$ with the same entities and connectivity structure as $\mathcal{G}$. Here, we make a distinction between state and state embedding. The state at $t$ is the actual KG after $t$ perturbation steps and is denoted as $\mathcal{G}_t$. The state embedding at $t$ is a vector representation of $\mathcal{G}_t$ and is denoted as $s_t$. To match $a_t$, we also decouple $s_t$ into *substate* embeddings: $s_t = (s_t^{(0)}, s_t^{(1)}, s_t^{(2)})$.

**Reward** The reward function pushes the policy to maximize downstream performance. For commonsense QA, higher reward corresponds to lower KL divergence between the predicted and true answer distributions. For item recommendation, we use validation AUC as the reward function.

### 4.2.1 DQN ARCHITECTURE AND TRAINING

As described above, RL-RR is modeled as an action-subaction hierarchy. At the action (top) level, for $t$, the policy selects an action $a_t$ given state $s_t$, then performs $a_t$ on $\mathcal{G}_t$ to obtain $\mathcal{G}_{t+1}$. At the subaction (bottom) level, for index $i \in [0, 1, 2]$ within time step $t$, the policy selects a subaction $a_t^{(i+1)}$ given $s_t^{(i)}$ and, if any, previous subactions.

At $t$, the policy takes as input the substate embedding $s_t^{(0)}$. One approach for computing $s_t^{(0)}$ would be to directly encode $\mathcal{G}_t$ with a graph encoder $\mathcal{F}^{\text{graph}}$, such that $s_t^{(0)} = \mathcal{F}^{\text{graph}}(\mathcal{G}_t)$ (Dai et al., 2018; Sun et al., 2020; Ma et al., 2019b). However, since we aim to assess graph encoders' ability to capture KG information, it would not make sense to use a graph encoder for KG perturbation. Instead, we use an LSTM (Hochreiter & Schmidhuber, 1997) to update substate embeddings both within and across time steps, while jointly encoding substate and subaction embeddings. Observe that

this means $s_t^{(i)}$ only implicitly captures KG state information via $a_t^{(i-1)}$, since the choice of each subaction is constrained precisely by which entities, relations, or edges are available in $\mathcal{G}_t$.

To train RL-RR, we use the DQN algorithm (Mnih et al., 2015). Abstractly, the goal of DQN is to learn a Q-function $Q(s_t, a_t)$, which outputs the expected reward for taking action $a_t$ in state $s_t$. In our implementation, $Q(s_t, a_t)$ is decomposed into a sequential pair of sub-Q-functions: $Q_1(a_t^{(1)}|s_t^{(0)}, a_t^{(0)}) = \langle \text{MLP}(a_t^{(1)}), \text{MLP}(h_t^{(0)}) \rangle$ and $Q_2(a_t^{(2)}|s_t^{(1)}, a_t^{(0)}, a_t^{(1)}) = \langle \text{MLP}(a_t^{(2)}), \text{MLP}(h_t^{(1)}) \rangle$. MLP denotes the vector representation computed by a multi-layer perceptron, while $h_t^{(0)}$ and $h_t^{(1)}$ denote the respective LSTM encodings of $(s_t^{(0)}, a_t^{(0)})$ and $(s_t^{(1)}, [a_t^{(0)} \oplus a_t^{(1)}])$.

Fig. 3 depicts the perturbation procedure at $t$. First, we either initialize $s_t^{(0)}$ with a trained embedding weight vector if $t=0$, or set it to $s_{t-1}^{(2)}$ otherwise. Second, we uniformly sample $a_t^{(0)}$, which is encoded as $h_t^{(0)} = \text{LSTMCell}_1(s_t^{(0)}, a_t^{(0)})$. LSTMCell$_1$ also updates $s_t^{(0)}$ to $s_t^{(1)}$. Third, we compute $Q_1(a_t^{(1)}|s_t^{(0)}, a_t^{(0)})$, which takes $h_t^{(0)}$ as input and outputs $a_t^{(1)}$. Fourth, we encode $a_t^{(1)}$ as $h_t^{(1)} = \text{LSTMCell}_2(s_t^{(1)}, [a_t^{(0)} \oplus a_t^{(1)}])$. LSTMCell$_2$ also updates

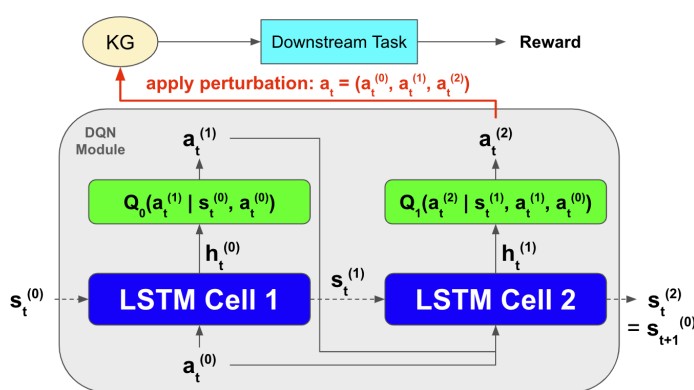

Figure 3: **DQN Architecture for RL-RR**

$s_t^{(1)}$ to $s_t^{(2)}$. Fifth, we compute $Q_2(a_t^{(2)}|s_t^{(1)}, a_t^{(0)}, a_t^{(1)})$, which takes $h_t^{(1)}$ as input and outputs $a_t^{(2)}$. Note that $a_t^{(1)}$ and $a_t^{(2)}$ are selected $\epsilon$-greedily during training and greedily during evaluation. Finally, using $a_t = (a_t^{(0)}, a_t^{(1)}, a_t^{(2)})$, we perturb $\mathcal{G}_t$ to get $\mathcal{G}_{t+1}$.

Ideally, for each $t$, we would evaluate $\mathcal{G}_{t+1}$ on the downstream task to obtain the reward. However, downstream evaluation is expensive, so we only compute reward every $T$ time steps. Moreover, for the policy to generalize well, state embeddings $(s_{t-T+1}, ..., s_{t-1}, s_t)$ should not correlate with the order of actions $(a_{t-T+1}, ..., a_{t-1}, a_t)$. Thus, for every $T$ time steps during training, we shuffle the last $T$ actions after computing reward, then update the LSTM and sub-Q-functions with respect to the shuffled actions. Doing so encourages state embeddings to be invariant to action order.

## 5 EXPERIMENTS

In this section, we test KG-augmented models on their ability to maintain performance and explainability when the KG has been extensively perturbed. As explained in Sec. 2 and Fig. 1, the model is first trained on a given dataset using the original KG, frozen throughout KG perturbation, then used to compare downstream performance between original KG and perturbed KG. For all models, datasets, and perturbation methods, we measure performance and KG similarity when all $|\mathcal{T}|$ KG edges have been perturbed, averaged over three runs. For a subset of model-dataset-perturbation configurations, we also measure performance as a function of the number of edges perturbed. In addition, we conduct a user study where humans are asked to rate original and perturbed KGs, with respect to readability and usability for solving downstream tasks.

### 5.1 COMMONSENSE QA

For commonsense QA, the KG-augmented models we experiment with are RN (with attentional path aggregation) (Lin et al., 2019; Santoro et al., 2017) and MHGRN (Feng et al., 2020), which have been shown to outperform non-KG models (Devlin et al., 2018; Liu et al., 2019) and a number of KG-augmented models (Lin et al., 2019; Ma et al., 2019a; Wang et al., 2019c; Schlichtkrull et al., 2018) on this task. For both RN and MHGRN, we use a BERT-Base (Devlin et al., 2018) text encoder. We evaluate on the CommonsenseQA (CSQA) (Talmor et al., 2018) and OpenBookQA (OBQA) (Mihaylov et al., 2018) datasets, using ConceptNet (Speer et al., 2016) as the KG. Performance is measured using accuracy (Acc), which is the standard metric for commonsense QA (Lin et al., 2019; Feng et al., 2020).

Table 1: **Comparison of Perturbation Methods on CSQA.** We follow the standard protocol of reporting CSQA test accuracy on the in-house data split from Lin et al. (2019), as official test labels are not available.

| | CSQA | | | | | | | |
|---|---|---|---|---|---|---|---|---|
| | RN | | | | MHGRN | | | |
| Method | Acc ($\uparrow$) | ATS ($\downarrow$) | SC2D ($\downarrow$) | SD2 ($\downarrow$) | Acc ($\uparrow$) | ATS ($\downarrow$) | SC2D ($\downarrow$) | SD2 ($\downarrow$) |
| No KG | 53.41 | - | - | - | 53.41 | - | - | - |
| Original KG | 56.87 | 0.940 | 1.000 | 1.000 | 57.21 | 0.940 | 1.000 | 1.000 |
| Relation Swapping (RS) | 53.42 | 0.831 | 0.144 | 6.16E-3 | 53.42 | 0.831 | 0.144 | 6.16E-3 |
| Relation Replacement (RR) | 53.42 | 0.329 | **0.091** | 1.70E-3 | 52.22 | 0.329 | **0.091** | **1.70E-3** |
| Edge Rewiring (ER) | 53.42 | 0.505 | 0.116 | 2.30E-3 | 52.22 | 0.505 | 0.116 | 2.30E-3 |
| Edge Deletion (ED) | 52.21 | 0.933 | 0.126 | 2.00E-3 | 51.00 | 0.933 | 0.126 | 2.00E-3 |
| RL-RR | **55.21** | **0.322** | 0.093 | **1.66E-3** | **55.52** | **0.314** | 0.092 | 1.78E-3 |

Table 2: **Comparison of Perturbation Methods on OBQA**

| | OBQA | | | | | | | |
|---|---|---|---|---|---|---|---|---|
| | RN | | | | MHGRN | | | |
| Method | Acc ($\uparrow$) | ATS ($\downarrow$) | SC2D ($\downarrow$) | SD2 ($\downarrow$) | Acc ($\uparrow$) | ATS ($\downarrow$) | SC2D ($\downarrow$) | SD2 ($\downarrow$) |
| No KG | 62.00 | - | - | - | 62.00 | - | - | - |
| Original KG | 66.80 | 0.934 | 1.000 | 1.000 | 68.00 | 0.934 | 1.000 | 1.000 |
| Relation Swapping (RS) | 67.00 | 0.857 | 0.159 | 7.73E-3 | 67.30 | 0.857 | 0.159 | 7.73E-3 |
| Relation Replacement (RR) | 66.80 | 0.269 | **0.095** | 1.84E-3 | 67.60 | 0.269 | 0.095 | 1.84E-3 |
| Edge Rewiring (ER) | 66.60 | 0.620 | 0.146 | 7.26E-3 | 67.00 | 0.620 | 0.146 | 7.26E-3 |
| Edge Deletion (ED) | 66.80 | 0.923 | 0.134 | 2.19E-3 | 67.60 | 0.923 | 0.134 | 2.19E-3 |
| RL-RR | **67.30** | **0.255** | 0.097 | **1.79E-4** | **67.70** | **0.248** | **0.094** | **1.75E-4** |

**CSQA** Results for CSQA are given in Table 1. For RN and MHGRN, we see that RL-RR achieves slightly worse accuracy than Original KG, while RS, RR, and ER perform on par with No KG. For both models, ED performs noticeably worse than No KG.

**OBQA** Results for OBQA are shown in Table 2. For RN, we see that RL-RR actually obtains better accuracy than Original KG. For MHGRN, RL-RR yields marginally worse accuracy than Original KG. Meanwhile, for both RN and MHGRN, all heuristics uniformly achieve similar accuracy as Original KG, which itself significantly outperforms No KG.

**Analysis** Tables 1-2 demonstrate that perturbing a KG does not necessarily imply decreased performance, nor does it guarantee the creation of invalid or novel facts. As shown by the KG similarity scores, some perturbation methods cause greater semantic or structural KG changes than others. Perturbed KGs produced by RS and ED have high ATS (i.e., semantic similarity to original KG), while RR, ER, and RL-RR achieve relatively low ATS. Meanwhile, SC2D and SD2 are quite low for all perturbation methods, indicating consistently low structural similarity between original and perturbed KG. RL-RR and RR collectively have the lowest SC2D and SD2 for CSQA, while RL-RR has the lowest SC2D and SD2 for OBQA. Notably, across all perturbation methods and models, RL-RR attains the highest accuracy while also having the lowest KG similarity scores overall. The results of a T-test (three runs for both models) show that RL-RR achieves a statistically significant improvement over its heuristic counterpart, RR. Still, even RR has a fairly high accuracy to KG similarity ratio. This suggests that our KG-augmented models are not using the KG in a human-like way, since RL-RR and RR can both achieve high performance despite extensively corrupting the original KG's semantic and structural information.

## 5.2 ITEM RECOMMENDATION

The KG-augmented recommender systems we consider are KGCN (Wang et al., 2019b) and RippleNet (Wang et al., 2018b). We evaluate these models on the Last.FM (Rendle, 2012) and MovieLens-20M (Harper & Konstan, 2016) datasets, using the item KG from Wang et al. (2019a). As mentioned in Sec. 1, item KGs have been shown to benefit recommender systems in cold start scenarios (Wang et al., 2018b). Therefore, following Wang et al. (2018b), we simulate a cold start scenario by using only 20% and 40% of the train set for Last.FM and Movie Lens-20M, respectively. Performance is measured using AUC, which is the standard metric for item recommendation (Wang et al., 2019b; 2018b). Since the item KG is almost bipartite, the local clustering coefficient of each item in the KG is extremely small, and so SC2D is not meaningful here (Sec. 3). Thus, for item recommendation, we do not report SC2D.

Table 3: **Comparison of Perturbation Methods on Last.FM**

| | Last.FM | | | | | |
| | KGCN | | | Ripplenet | | |
| Method | AUC (↑) | ATS (↓) | SD2 (↓) | AUC (↑) | ATS (↓) | SD2 (↓) |
|---|---|---|---|---|---|---|
| No KG | 50.75 | - | - | 50.75 | - | - |
| Original KG | 55.99 | 0.972 | 1.000 | 56.23 | 0.972 | 1.000 |
| Relation Swapping (RS) | 55.98 | 0.681 | 2.96E-2 | 56.23 | 0.681 | 2.96E-2 |
| Relation Replacement (RR) | 55.98 | 0.415 | 0.253 | 56.22 | 0.415 | 0.253 |
| Edge Rewiring (ER) | 55.98 | 0.437 | 1.85E-2 | 53.74 | 0.437 | 1.85E-2 |
| Edge Deletion (ED) | 50.96 | 0.941 | 3.37E-3 | 45.98 | 0.941 | 3.37E-3 |
| RL-RR | **56.04** | **0.320** | **1.30E-3** | **56.28** | **0.310** | **1.20E-3** |

Table 4: **Comparison of Perturbation Methods on MovieLens-20M**

| | MovieLens-20M | | | | | |
| | KGCN | | | Ripplenet | | |
| Method | AUC (↑) | ATS (↓) | SD2 (↓) | AUC (↑) | ATS (↓) | SD2 (↓) |
|---|---|---|---|---|---|---|
| No KG | 91.30 | - | - | 91.30 | - | - |
| Original KG | 96.62 | 0.960 | 1.000 | 97.46 | 0.960 | 1.000 |
| Relation Swapping (RS) | **96.62** | 0.678 | 6.14E-4 | 97.46 | 0.678 | 6.14E-4 |
| Relation Replacement (RR) | 96.50 | 0.413 | **7.74E-5** | 97.45 | 0.413 | **7.74E-5** |
| Edge Rewiring (ER) | 96.24 | 0.679 | 4.44E-4 | 93.42 | 0.679 | 4.44E-4 |
| Edge Deletion (ED) | 90.36 | 0.982 | 1.02E-4 | 90.22 | 0.982 | 1.02E-4 |
| RL-RR | 96.53 | **0.401** | 2.23E-4 | 97.25 | **0.268** | 2.21E-4 |

**Last.FM** Results for Last.FM are shown in Table 3. For KGCN and RippleNet, we see that RS, RR, and RL-RR achieve about the same AUC as Original KG, with RL-RR slightly outperforming Original KG. ER performs similarly to Original KG for KGCN, but considerably worse for RippleNet. ED's AUC is on par with No KG's for KGCN and much lower than No KG's for RippleNet.

**MovieLens-20M** Results for MovieLens-20M are displayed in Table 4. For both KGCN and RippleNet, we find that relation-based perturbation methods tend to perform on par with Original KG. Here, ER is the better of the two edge-based perturbation methods, performing about the same as Original KG for KGCN, but noticeably worse for RippleNet. Somehow, for both KGCN and RippleNet, ED achieves even worse AUC than No KG. On the other hand, we see that ED achieves very high ATS, while RS, RR, ER, and RL-RR achieve more modest ATS scores.

**Analysis** Like in commonsense QA, Tables 3-4 show that KG-augmented models can perform well even when the KG has been drastically perturbed. Using the T-test with three runs, for almost all perturbation methods, we find a statistically insignificant difference between the perturbed KG's AUC and the original KG's AUC. The perturbed KGs produced by ED have high ATS, while RS, RR, ER, and RL-RR achieve modest ATS scores. However, all perturbation methods have fairly low SD2 (except RR on Last.FM). In particular, across both datasets and models, RL-RR has the highest AUC overall, while also having the lowest KG similarity scores overall. This serves as additional evidence that the model is not using the KG in a human-like manner, since RL-RR achieves high performance despite significantly perturbing the original KG's semantic and structural information.

## 5.3 AUXILIARY EXPERIMENTS AND ANALYSIS

**Varying Perturbation Level** For a subset of model-dataset-perturbation settings, we measure the performance and ATS of various perturbation methods as a function of the percentage of KG edges perturbed. For MHGRN on CSQA, Fig. 4a shows that, across all levels of perturbation, RL-RR maintains higher accuracy than No KG. Meanwhile, RS's accuracy reaches No KG's accuracy at 100% perturbation, and RR's does so at 60% perturbation. In Fig. 4b, we see that RL-RR's and RR's ATS drop significantly as the perturbation percentage increases, whereas RS's ATS remains quite high even at 100% perturbation. For RippleNet on MovieLens-20M, Fig. 4c shows a flat performance curve for all perturbation methods. Meanwhile, for all perturbation methods in Fig. 4d, ATS decreases steadily as perturbation level increases, with RL-RR's ATS dropping most.

These findings support the hypothesis that KG perturbation does not imply performance decrease or KG corruption. Building on the results of previous experiments, in both model-dataset settings, RL-RR largely maintains the model's performance despite also heavily perturbing the KG's seman-

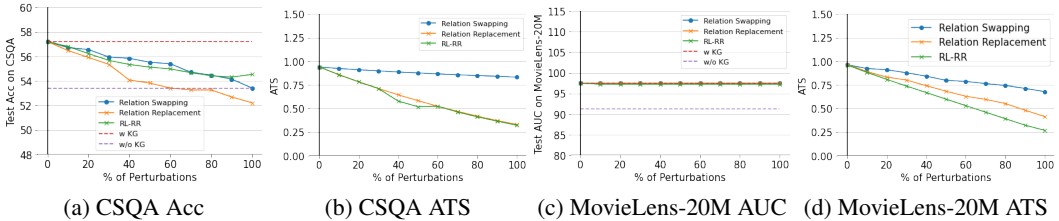

| (a) CSQA Acc | (b) CSQA ATS | (c) MovieLens-20M AUC | (d) MovieLens-20M ATS |

Figure 4: **Varying Perturbation Level.** Performance and ATS with respect to perturbation level, for MHGRN on CSQA and RippleNet on MovieLens-20M. Horizontal axis denotes percentage of perturbed KG edges.

| Method | CSQA | | OBQA | |
|---|---|---|---|---|
| | RN | MHGRN | RN | MHGRN |
| No KG | 53.41 | 53.41 | 62.00 | 62.00 |
| Orignal KG | 56.87 | 57.21 | 66.80 | 68.00 |
| Zero Subgraph Emb. | 53.10 | 53.99 | 64.80 | 66.40 |
| Rand. Subgraph Emb. | 52.60 | 52.48 | 64.75 | 65.90 |
| Rand. Ent./Rel. Emb. | 53.02 | 54.03 | 64.45 | 64.85 |

Table 5: **Noisy Baselines for Commonsense QA.** Noisy baseline accuracy on CSQA and OBQA.

| Method | Last.FM | | MovieLens-20M | |
|---|---|---|---|---|
| | KGCN | RippleNet | KGCN | RippleNet |
| No KG | 50.75 | 50.75 | 91.30 | 91.30 |
| Original KG | 55.99 | 56.23 | 96.62 | 97.46 |
| Rand. Ngbd. | 55.91 | 51.04 | 96.21 | 92.11 |

Table 6: **Noisy Baselines for Item Recommendation.** Noisy baseline AUC on Last.FM and MovieLens-20M.

tics. Interestingly, for RippleNet on MovieLens-20M, performance is completely unaffected by KG perturbation, even though the KG's semantic information is apparently being corrupted.

**Noisy Baselines** To see if KGs yielded by our perturbation methods capture more than just random noise, we compare them to several noisy baselines. Table 5 gives results for three noisy baselines on commonsense QA: (1) replace subgraph embedding with zero vector, (2) replace subgraph embedding with random vector, and (3) replace entity/relation embeddings with random vectors. For CSQA, the noisy baselines perform noticeably worse than both Original KG and RL-RR, while being on par with No KG (Table 1). For OBQA, the noisy baselines' perform slightly better than No KG, but considerably worse than Original KG and all of the perturbation methods (Table 2). Table 6 displays results for our noisy baseline in item recommendation, which entails randomizing each entity's neighborhood. We find that KGCN performs about the same for this noisy baseline as for Original KG and our best perturbation methods, whereas RippleNet performs much worse (Tables 3-4). RippleNet may be more sensitive than KGCN to entity neighbor randomization because RippleNet considers directed edges. This is supported by RippleNet's performance dropping when we perturb edge connections (Tables 3-4). In both tasks, the noisy baselines show that our perturbation methods yield KGs that capture measurably useful information beyond just noise. For KGCN, the unexpected discovery that noisy baselines perform similarly to Original KG suggest that even noisy KGs can contain useful information for KG-augmented models.

**Human Evaluation of KG Explanations** We conduct a user study to measure the plausibility of KG-augmented models' path-based explanations. For both the original KG and RL-RR perturbed KG, we sample 30 questions from the CSQA and OBQA test sets which were correctly answered by MHGRN. For each question, we retrieve the top-scoring path for each answer choice via MHGRN's path decoder attention. We then ask three human subjects to rate each path for readability and usability, with ratings aggregated via majority voting. Readability (Read) is whether the path

| Method | CSQA | | OBQA | |
|---|---|---|---|---|
| | Read | Use | Read | Use |
| Orig. KG | 0.360 | 0.081 | 0.357 | 0.111 |
| RL-RR | 0.353 | 0.115 | 0.199 | 0.100 |

Table 7: **Human Evaluation of KG Explanations.** Human ratings for readability (Read) and usability (Use) of KG explanation paths, on a $[0, 1]$ scale.

makes sense, usability (Use) is whether the path is relevant to the given question-answer pair, and both are measured on a $[0, 1]$ scale. We obtain a Fleiss' $\kappa$ of $0.1891$, indicating slight agreement between raters. To illustrate, we provide examples of explanation paths and their consensus ratings. Given the question *James chose to not to print the cards, because he wanted to be more personal. What type of cards did he choose, instead?*, the Original KG path is PRINT —[ANTONYM]→ HANDWRITTEN (Read=1.0; Use=2.0), and the RL-RR path is PRINT — [NOTDESIRES]→ HANDWRITTEN (Read=0.0; Use=0.0). Here, the Original KG path seems plausible, but the RL-RR path does not.

In Table 7, we see that Original KG and RL-RR got relatively low ratings for both readability and usability. Whereas MHGRN successfully utilizes all KG paths in this user study, humans largely struggle to read or use them. This suggests that KG-augmented models and humans process KG information quite differently, thus challenging the role of KG paths as plausible explanations. Also, Original KG beats RL-RR in readability and usability overall, signaling RL-RR's ability to corrupt the KG. CSQA's lower sensitivity to perturbation can be explained by the fact that CSQA is con-

structed from ConceptNet. Every CSQA question-answer is based on ConceptNet entities/relations, so a random ConceptNet subgraph is more likely to have semantic overlap with a CSQA question-answer than with an OBQA question-answer. Hence, a perturbed ConceptNet subgraph may also be more likely to overlap with a CSQA question-answer, which means perturbing the KG might have a smaller impact on human judgments of CSQA paths. Note that this result concerns explainability and does not say anything about the model's performance on CSQA and OBQA.

**Validation of KG Similarity Metrics** Using our human evaluation results, we validate our three proposed KG similarity metrics: ATS, SC2D and SD2. We find that the Pearson correlation coefficient between the human evaluation scores in Table 7 and the three KG similarity scores in Tables 1-2 are 0.845, 0.932 and 0.932, respectively. This indicates high correlation and that our metrics aptly capture a perturbed KG's preservation of semantic/structural information from its original KG.

**Why do perturbed KGs sometimes perform better than the original KG?** In our experiments, relation-based perturbations (RS, RR, RL-RR) generally outperform edge-based perturbations (ER, ED). Also, we find that the original KG can contain noisy relation annotations which are sometimes "corrected" by relation-based perturbations. In certain cases, this may result in the perturbed KG achieving slightly higher performance than the original KG (RR and RL-RR for RN-CSQA; RL-RR for Last.FM). Similarly, in our user study, despite all questions being correctly answered by the model, there were some RL-RR explanations that received higher readability/usability ratings than their original KG counterparts. Although the original KG achieved higher human ratings than the RL-RR KG did overall, both KGs still achieved relatively low ratings with respect to our scales. While our main argument centers on KG-augmented models' flaws, this counterintuitive finding suggests that KGs themselves are flawed too, but in a way that can be systematically corrected.

## 6 RELATED WORK

**KG-Augmented Neural Models** Although neural models may already capture some semantic knowledge (Petroni et al., 2019; Davison et al., 2019), augmenting them with external KGs has improved performance on various downstream tasks: commonsense QA (Lin et al., 2019; Shen et al., 2020; Lv et al., 2020; Musa et al., 2019), item recommendation (Wang et al., 2019b; 2020; Song et al., 2019; Cao et al., 2019), natural language inference (Chen et al., 2017; Wang et al., 2019c), and others (Chen et al., 2019; Kapanipathi et al.). KG-augmented models have also been designed to explain the model's predictions via attention over the KG (Lin et al., 2019; Zhang et al., 2019; Song et al., 2019; Cao et al., 2019; Gao et al., 2019; Ai et al., 2018).

**Adversarial Perturbation of Graphs** Inspired by adversarial learning in computer vision (Bhambri et al., 2019) and NLP (Zhang et al., 2020), some recent works have addressed adversarial perturbation in graph learning (Chen et al., 2020). Multiple paradigms have been proposed for graph perturbation, including gradient-based methods (Chen et al., 2018; Bojchevski & Günnemann, 2019; Wu et al., 2019), RL-based methods (Ma et al., 2019b; Dai et al., 2018), and autoencoder-based methods (Chen et al., 2018). Whereas such works aim to minimally perturb the graph while maximally impacting the graph's performance, our purpose for graph perturbation is to see whether KG-augmented models' use KGs in a human-like way and provide plausible explanations.

## 7 CONCLUSION

In this paper, we analyze the effects of strategically perturbed KGs on KG-augmented model predictions. Using four heuristics and a RL policy, we show that KGs can be perturbed in way that drastically changes their semantics and structure, while preserving the model's downstream performance. Apparently, KG-augmented models can process KG information in a way that does not align with human priors about KGs, although the nature of this process still requires further investigation. Moreover, we conduct a user study to demonstrate that both perturbed and unperturbed KGs struggle to facilitate plausible explanations of the model's predictions. Note that our proposed KG perturbation methods merely serve as analytical tools and are not intended to directly improve model performance or explainability. Nonetheless, we believe our findings can guide future work on designing KG-augmented models that are better in these aspects. Additionally, our results suggest that KG-augmented models can be robust to noisy KG data. Even when the KG contains a fairly small amount of signal, these models are somehow able to leverage it. This could be useful in situations where it is impractical to obtain fully clean KG annotations.

## 8 ACKNOWLEDGMENTS

This research is supported in part by the Office of the Director of National Intelligence (ODNI), Intelligence Advanced Research Projects Activity (IARPA), via Contract No. 2019-19051600007, the DARPA MCS program under Contract No. N660011924033 with the United States Office Of Naval Research, the Defense Advanced Research Projects Agency with award W911NF-19-20271, and NSF SMA 18-29268. We would like to thank all collaborators at the USC INK Research Lab for their constructive feedback on the work.

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
