# OpenReview forum: "Learning to Deceive Knowledge Graph Augmented Models via Targeted Perturbation"
_ICLR.cc/2021/Conference — ICLR 2021 Poster_

### Official Review · AnonReviewer3 · 2020-10-27
**An interesting paper with interesting findings**

**Rating:** 6
**Confidence:** 3

**Review:**

The paper presents an interesting finding that some of the existing KG-augmented models, such as those for QA and item recommendation, may not actually capture or leverage the semantics in KGs, and their performance improvement cannot be attributed to the usage of additional knowledge. I think this finding is of some significance.

Pros:

1) The paper presents four heuristic methods and an RL-based method for KG perturbation. It obtains some interesting and noteworthy findings based on the experimental results in terms of QA and recommender systems.

Cons:

1) In my opinion, perturbing KGs by randomly or heuristically changing existing edges is not well-motivated. Although it supports the experimental study in this paper and reveals several interesting findings, it has very little practical significance, because the perturbed KG would inevitably contain many invalid and incorrect facts without positive effects for downstream applications. Hence, the technical contribution of the proposed heuristic strategies and RL model for KG perturbation is not significant.

2) The analysis for experimental results is somewhat superficial. It fails to provide deep insights into why some models can still work with the perturbed KG. The authors should provide more analysis to explain the experimental results that are against common sense. For example, an experiment conclusion says that "it is the (false) connection between entities instead of the semantic stored in the KG that leads to the improvement over non-KG baseline." Why? I think this is an interesting and noteworthy finding, but no in-depth analysis is given. Besides, I also have a minor question, i.e., how does the edge deletion method perturb all the triples (100%) in a KG? How many triples are left after edge deletion?

Some typos:

1) these false connection ->  these false connections
2) Relation Swapping(RS) -> Relation Swapping (RS)

Anyway, the paper presents a noteworthy finding, which calls for further investigation into what information from the KGs is actually captured to improve the neural-symbolic models. So, I would like to recommend a weak acceptance of this paper.

--- after rebuttal ---
Thank the authors for their response which addressed my concerns. Based on the response, the revision, and other reviews, I would like to keep my score unchanged at 6.

---

> ### Author Response · Authors · 2020-11-19
> **Response to AnonReviewer3**
>
> Thank you for your feedback! Below are responses to your comments/questions.
>
> -----
>
> **3.1: In my opinion, perturbing KGs by randomly or heuristically changing existing edges is not well-motivated. Although it supports the experimental study in this paper and reveals several interesting findings, it has very little practical significance, because the perturbed KG would inevitably contain many invalid and incorrect facts without positive effects for downstream applications. Hence, the technical contribution of the proposed heuristic strategies and RL model for KG perturbation is not significant.**
>
> By showing that KG-augmented neural-symbolic models perform well on perturbed KGs, we disprove existing assumptions about how models use KG info and about the plausibility of explanations provided by such models. Contrary to popular belief, KG-augmented models can still work even when the KG has been perturbed so much that humans cannot understand the models’ explanations. In light of this, the motivation of our paper is to guide future work on designing models that use KG info effectively and provide plausible explanations. Note that the proposed KG perturbation methods merely serve as analytical tools and are not intended to directly improve performance or explanation quality. Additionally, our results suggest that KG-augmented neural-symbolic models can be robust to noisy KG data. Even when the KG contains a fairly small amount of signal, the models are somehow able to leverage it. This could be a useful property in situations where it is not practical to obtain fully clean KG annotations.
>
> We have updated our paper to explicitly describe this motivation. Please refer to *Section 7: Conclusion*.
>
>
> **3.2: The analysis for experimental results is somewhat superficial. It fails to provide deep insights into why some models can still work with the perturbed KG. The authors should provide more analysis to explain the experimental results that are against common sense. For example, an experiment conclusion says that "it is the (false) connection between entities instead of the semantic stored in the KG that leads to the improvement over non-KG baseline." Why? I think this is an interesting and noteworthy finding, but no in-depth analysis is given.**
>
> Good question!
>
> First, subgraph extraction is heuristic-based, so some relevant info in the original KG may not be retrievable unless the KG itself is changed. Meanwhile, the RL-based KG perturbation methods are trained to maximize performance, which may enable the KG to be modified such that more relevant info is retrievable using the subgraph extraction heuristics.
>
> Second, in our experiments, relation-based perturbations (RS, RR, RL-RR) generally outperform edge-based perturbations (ER, ED). Also, we have found that the original KG can contain noisy relation annotations which are sometimes “corrected” by relation-based perturbations. In certain cases, this may result in the perturbed KG achieving slightly higher performance than the original KG (RR and RL-RR for RN-CSQA; RL-RR for Last.FM). Similarly, in our user study, despite all questions being correctly answered by the model, there were some RL-RR explanations that received higher readability/usability ratings than their original KG counterparts. Although the original KG achieved higher human ratings than the RL-RR KG did overall, both KGs still achieved relatively low ratings with respect to our scales. While our main argument centers on NSKG models' flaws, this counterintuitive finding suggests that KGs themselves are flawed too, but in a way that can be systematically corrected.
>
> We’ve updated the paper to include this analysis, which we agree would make our claims more convincing. Please refer to *Section 5.3: Auxiliary Experiments and Analysis -- Paragraph “Why do perturbed KGs sometimes perform better than the original KG?”*.
>
>
> **3.3: Besides, I also have a minor question, i.e., how does the edge deletion method perturb all the triples (100%) in a KG? How many triples are left after edge deletion?**
>
> Good catch. For edge deletion, at 100% perturbation, we delete all but 10 edges.
>
> We’ve updated the paper to include this info. Please refer to *Section 3.1: Heuristic-Based KG Perturbation*.

---

### Official Review · AnonReviewer1 · 2020-10-28
**This paper is not properly motivated or need better justification.**

**Rating:** 4
**Confidence:** 4

**Review:**

This paper proposed to learn a RL model to modified the KG. They showed that their model can successfully deceive the KG-augmented models with most relations replaced, compared to heuristic-based strategies.

I am not convinced by the motivation of this paper. The authors claimed that their model can learn to modified a KB so that a KG-augmented model can yield similar performance as before. This is under the assumption that similar performance in predicting the correct answer (e.g. accuracy) will lead to similar quality of explanation. This assumption does not always hold. If the authors really care about explanation, you should experiment with the explanation model that used KGs.

There's also a discrepancy between the authors' motivation and their evaluation. If the authors assume KG is really important in generating explanations and can be easily fooled by their framework, they should also evaluated on KG only tasks, e.g. KBQA, KBC etc, besides these KG-augmented models.

Some detailed questions:
1. Many of the downstream models that the authors evaluated on are based on GCN approaches. GCN is good at filtering irrelevant information from their neighbors. How did you make sure that embeddings from the replaced relations are not simply filtered out at the aggregation steps?
2. I would like to argue the LSTM is not appropriate here. LSTM is designed for "ordered" input. Making tricks to adapt it to un-ordered inputs is not a very good option. In each update step, the LSTM basically learns a gate variable \sigma on the input. Can you do something like: G += \sum \alpha \dot g, where g is the embedding of your single updates?
3. In some cases, with replaced relations, the model can have better performance as the original model w/ KG (e.g. OBQA RN Acc in Table 1). Do you have an intuition why that could happen?

---

> ### Author Response · Authors · 2020-11-19
> **Response to AnonReviewer1 (Part 1)**
>
> Thank you for your feedback! Below, we’ve clarified our motivation and why we believe this motivation is aligned with our experiments. Also, we’ve provided responses to the three questions you listed.
>
> -----
>
> **1.1: Clarification about motivation**
>
> We would like to clarify what we think may be a fundamental misunderstanding here. In this paper, we demonstrate that KG-augmented models can still maintain their downstream performance even when the KG has been perturbed significantly. Our motivation for doing so is to show that high model performance does NOT imply high explanation faithfulness. In other words, we do NOT assume that “similar performance in predicting the correct answer (e.g., accuracy) will lead to similar quality of explanation”, but show the opposite instead. Note that all of the models we evaluated were claimed to provide KG-based explanations for their predictions.
>
> The surprising fact that KG-augmented models can still work on perturbed KGs raises doubts about how these models actually use KG info. As demonstrated in the human evaluation, such models apparently use KG info in a way that is not aligned with how humans interpret their explanations. We believe our findings are an important step in guiding future work on designing models that more effectively use KG info to both improve performance and provide better explanations.
>
>
> **1.2: Clarification about experiments**
>
> We believe that our experiments support the motivation described in 1.1. As suggested, KG-only tasks could also be a reasonable setting. However, we specifically consider KG-augmented tasks because they naturally provide a non-KG baseline to compare against (i.e., the “No KG” results). Since our goal is to demonstrate that a perturbed KG can still be useful to neural-symbolic models, we need to be able to show that using a perturbed KG is better than not using a KG at all.
>
>
> **1.3: Many of the downstream models that the authors evaluated on are based on GCN approaches. GCN is good at filtering irrelevant information from their neighbors. How did you make sure that embeddings from the replaced relations are not simply filtered out at the aggregation steps?**
>
> Given a fixed model, our goal is to measure how well the model performs on a perturbed KG, compared to on the original KG. Thus, if GCN is good at filtering irrelevant information from node neighbors, we would like to evaluate this ability on both original and perturbed KGs. To facilitate a fair comparison, we feel it would be best not to handicap the GCN to perform worse on the perturbed KG by “mak[ing] sure that embeddings from the replaced relations are not simply filtered out at the aggregation steps”. Nonetheless, we agree that the effects of perturbed KGs on GCN’s internal mechanisms should be further analyzed in future work (e.g., at each layer, which KG facts are being emphasized?).

---

> > ### Author Response · Authors · 2020-11-19
> > **Response to AnonReviewer1 (Part 2)**
> >
> > **1.4: I would like to argue the LSTM is not appropriate here. LSTM is designed for "ordered" input. Making tricks to adapt it to un-ordered inputs is not a very good option. In each update step, the LSTM basically learns a gate variable \sigma on the input. Can you do something like: G += \sum \alpha \dot g, where g is the embedding of your single updates?**
> >
> > We appreciate your feedback on this tricky design choice! We did also consider a state update function like the one you suggested. Concretely, to compute the state embedding at time step t, the function we considered was: s_t = \mean(F(a_0), F(a_1), ... F(a_{t-1})) = \sum_i (1/t) * F(a_i), where F is an encoder and a_i is the action embedding at time step i. Although this state update function strictly preserves order invariance, it is not ideal because s_t approaches constant value as t becomes large.
> >
> > Now, let’s denote the LSTM state update function with action shuffling as “LSTM-Shuffle”. Concretely, for every T time steps during training, we shuffle the last T actions after computing reward, then update the LSTM and sub-Q-functions with respect to the shuffled actions. Unlike the previous mean-based state update function, LSTM-Shuffle does not suffer from decaying state updates as t becomes large. Meanwhile, LSTM-Shuffle does decorrelate state embeddings with action order, albeit only approximately. Across different permutations of a given action sequence, only the DQN reward (not the states) is pushed to be the same. Thus, since LSTM-Shuffle’s state embeddings are not guaranteed to be the same for different action sequence permutations, we cannot consider its states to be strictly order-invariant.
> >
> >
> > **1.5: In some cases, with replaced relations, the model can have better performance as the original model w/ KG (e.g. OBQA RN Acc in Table 1). Do you have an intuition why that could happen?**
> >
> > Good question!
> >
> > First, subgraph extraction is heuristic-based, so some relevant info in the original KG may not be retrievable unless the KG itself is changed. Meanwhile, the RL-based KG perturbation methods are trained to maximize performance, which may enable the KG to be modified such that more relevant info is retrievable using the subgraph extraction heuristics.
> >
> > Second, in our experiments, relation-based perturbations (RS, RR, RL-RR) generally outperform edge-based perturbations (ER, ED). Also, we have found that the original KG can contain noisy relation annotations which are sometimes “corrected” by relation-based perturbations. In certain cases, this may result in the perturbed KG achieving slightly higher performance than the original KG (RR and RL-RR for RN-CSQA; RL-RR for Last.FM). Similarly, in our user study, despite all questions being correctly answered by the model, there were some RL-RR explanations that received higher readability/usability ratings than their original KG counterparts. Although the original KG achieved higher human ratings than the RL-RR KG did overall, both KGs still achieved relatively low ratings with respect to our scales. While our main argument centers on NSKG models' flaws, this counterintuitive finding suggests that KGs themselves are flawed too, but in a way that can be systematically corrected.
> >
> > We’ve updated the paper to include this analysis, which we agree would make our claims more convincing. Please refer to *Section 5.3: Auxiliary Experiments and Analysis -- Paragraph “Why do perturbed KGs sometimes perform better than the original KG?”*.

---

### Official Review · AnonReviewer4 · 2020-10-28
**Great work on an interesting problem**

**Rating:** 7
**Confidence:** 3

**Review:**

This paper shows that knowledge graph augmented question answering and recommendation system models are so graph structure/content change invariant that by using a simple heuristic or more sophisticated reinforcement learning-based approach, we can change the knowledge graphs without significant change in the model performance. This phenomenon leads to corrupt non-sense explanations for such models' decisions/outputs.

Strengths:
- The paper is very well-written and crystal clear.
- The idea is very interesting and novel.
- The evaluations are relatively strong.

Weak points:
- I suggest the authors make their arguments about the invalidity of model explanations more clear by providing some explanation samples.
- I am wondering what is the source of a more significant gap in the VALIDITY of explanations for original kg vs RL-PR in OBQA compared to CSQA. I am suggesting the authors provide an explanation for this difference.

Overall, a study on the behavior of knowledge graph augmented models when they encounter the perturbed graphs is an interesting idea. The paper is also very well-written. I'd like to see the revised manuscript being accepted by the conference.

---

> ### Author Response · Authors · 2020-11-19
> **Response to AnonReviewer 4**
>
> Thank you for your feedback! Below are responses to the two concerns you expressed.
>
> ----------
>
> **4.1: I suggest the authors make their arguments about the invalidity of model explanations more clear by providing some explanation samples.**
>
> We agree that providing concrete examples of faithful and unfaithful explanations would improve our claims. Following your suggestion, In Figure X of the updated paper, we further illustrate our human evaluation by including real examples of fact path explanations produced by MHGRN using the original KG and RL-RR on CSQA, along with the consensus user ratings given to each explanation.
>
> Below is the example we’ve added to the paper:
>
> Example 1: Original KG seems plausible. RL-RR does not seem plausible.
> - Question: “James chose to not to print the cards, because he wanted to be more personal. What type of cards did he choose, instead?” (Answer: “handwritten”)
> - Original KG: “print ---[Antonym]---> handwritten” (Read: 1.0; Use: 2.0)
> - RL-RR: “print ---[NotDesires]---> handwritten” (Read: 0.0; Use: 0.0)
>
> Please refer to *Section 5.3: Auxiliary Experiments and Analysis -- Paragraph “Human Evaluation of KG Explanations”*.
>
> For reference, we also provide the second example below (not in the paper):
>
> Example 2: Original KG and RL-RR both do not seem plausible.
> - Question: “The middle of the day usually involves the bright star nearest to the earth to be straight overhead why?” (Answer: “human planet rotation”)
> - Original KG: “wind ---[IsA]---> rotation” (Read: 0.2; Use: 0.0)
> - RL-RR: “straight <--[NotCapableOf]--- turning ---[NotDesires]---> rotation” (Read: 0.4; Use: 0.0)
>
>
>
>
> **4.2: I am wondering what is the source of a more significant gap in the VALIDITY of explanations for original kg vs RL-RR in OBQA compared to CSQA. I am suggesting the authors provide an explanation for this difference.**
>
> Good question! First, note that we’ve renamed the user study dimensions as follows: [Interpretability --> Readability] and [Validity --> Usability].
>
> OBQA's much larger usability gap between Original KG and RL-RR can be explained by the fact that CSQA is constructed from ConceptNet. Every CSQA question-answer is based on ConceptNet entities/relations, so a random ConceptNet subgraph is more likely to have semantic overlap with a CSQA question-answer than with an OBQA question-answer. Hence, a perturbed ConceptNet subgraph may also be more likely to overlap with a CSQA question-answer, and so perturbing the KG might have a smaller impact on human judgments of CSQA path usability. This does not say anything about the model’s performance on CSQA and OBQA, as high model performance does not necessarily imply high explanation quality. In fact, our user study disproves this connection.
>
> We’ve updated our paper to include this analysis.
>
> Please refer to *Section 5.3: Auxiliary Experiments and Analysis -- Paragraph “Human Evaluation of KG Explanations”*.

---

### Official Review · AnonReviewer2 · 2020-10-29
**Review for Learning to Deceive Knowledge Graph Augmented Models via Targeted Perturbation**

**Rating:** 6
**Confidence:** 4

**Review:**

The paper provides a number of adversarial attacks on hybrid neural-symbolic systems. The systems are recommender and QA systems which use an underlying knowledge-graph (KG) such as ConceptNet. Previous work has suggested that the KGs are important for good performance, and moreover that the use of KGs lends the system a degree of interpretability. The attacks are successful - maintaining performance whilst seriously degrading the KG - throwing doubt on these claims.

Two main approaches to attacking the systems are followed: a simple heuristic method in which labels in the KG are modified randomly, and a more sophisticated method in which deep reinforcement learning is used to learn an optimal policy to change the KG whilst maintaining good performance on the task.

Overall I have a lot of sympathy for the motivation of this paper. There is increasing evidence that the attempted use of symbolic methods in hybrid systems, and in particular the use of symbolic methods to provide explanations, is just picking up on incidental properties of the data and exploiting the power of the deep network in ways unrelated to the symbolic representations. This paper provides further compelling evidence.

However, the paper is currently an extremely frustrating read, and it took me a number of attempts to get through the paper to write this review. Part of the problem is that the authors have tried to cram in too much material. I would have preferred to have seen fewer experiments described, but in more detail, with the remainder briefly mentioned in a paragraph or two, or perhaps in an appendix (if that's allowed for ICLR).

The other problem is that the presentation in the paper is poor. Part of that is down to the non-native English in parts (which is not the fault of the authors), but part of it is also just down to sloppiness in the presentation.

More detailed comments
--

A KG is denoted as G = (E, R, T ), where E and R are the entity set
and relation set respectively - this is a little confusing since T is
also the set of relations (tuples over E). R is the set of relation
*labels*?

Finally, both c and k are concatenated for calculating the
 plausibility score - [presuambly the concatenation is then put
 through an MLP?]

In both graph encoders, the subgraph G(q,a) together with the
aggregation weights - [neither of the descriptions mention aggregation
weights]

while the task is to predict the unobserved interaction (yuv = 0). -
[does the zero here mean that the interaction is unobserved, or that
the user has not engaged with the item? (or both?)]

where the aggregation weights are personalized for u - [you need to
define how the aggregation works]

We randomly choose two triples (edges) in the KG and swap their
relations. - [it would be good to see some actual examples here from
one of the applications/KGs.]

where sG(·) is a KG scoring function trained on G.  - [is this defined
anywhere?]

The font in table 1 is really small. If you get more room in a final
version please turn this into two tables.

in-house test accuracy - what's an "in-house" accuracy?

We then compute the relation
specific clustering coefficient vector c^r - do you define this anywhere?

OpenBookQA (OBQA) - do you give a reference?

We randomly select 10 questions - this seems like a small sample on
which to peform the human evaluation. This evaluation would be more
persuasive with a larger sample (even 20 or 30).

Typos etc. (not exhaustive)
--

despite a deceptive symbolic structure - [I'm not sure that
"deceptive" is the right word here (and lots of places elsewhere),
just something like "incorrect" may be better.]

 that KG -> the KG

The preliminary results indicate that KG can be easily manipulated and
lost its benefit -> lose

It brings more worrisome scenario - [rephrase]

without noticeable performance drop -> without a noticeable
performance drop

In specific -> More specifically

e.g., commonsense question answering (QA) and recommender system, ->
i.e.

aggregating its neighbors’ embedding -> aggregating its neighbors’ embeddings

This heuristic changes the semantic -> semantics

This heuristic also does not perturb KG’s structure but its semantic
-> This heuristic also does not perturb KG’s structure but its
semantics

on both of commonsense QA and recommendation system tasks -> on both
 commonsense QA and recommendation system tasks

since one
of our goal -> goals

have captured the information of KG. -> the KG

sequentially to LSTM -> sequentially to the LSTM

Evaluating the KG on downstream task -> Evaluating the KG on a
downstream task

Relation Swapping(RS) -> Relation Swapping (RS)

but keep the relation distribution -> keeps

We also leverage the validity scores given by human -> humans

ranging from gradient based(Chen - [space]

AutoEncoder based(Chen - [space]

which can lead to corrupt explanations -> which can lead to incorrect
explanations

our RL-RR method always yield -> yields

between entities instead of the semantic -> semantics

that investigate into the problem -> that investigate the problem

---

> ### Author Response · Authors · 2020-11-19
> **Response to AnonReviewer2 (Part 1)**
>
> Thank you for your feedback! Below are responses to each of your comments. Also, we’ve made major updates to the paper, incorporating your suggestions.
>
> -----
>
> **2.1: A KG is denoted as G = (E, R, T ), where E and R are the entity set and relation set respectively - this is a little confusing since T is also the set of relations (tuples over E). R is the set of relation labels?**
>
> Yes, R is the set of relation labels (a.k.a. relation types), while T is the set of facts (a.k.a. edges, triples), which are of the form (entity1, relation label, entity2). In ConceptNet, R contains relation labels such as “AtLocation” and “Desires”, while T contains fact triples like “(Desk, AtLocation, Classroom)” and “(Child, Desires, Learn)”.
>
> We have clarified this in the updated paper. Please refer to *Section 2: Problem Setting -- Paragraph "Notation"*.
>
>
>
> **2.2: Finally, both c and k are concatenated for calculating the plausibility score - [presumably the concatenation is then put through an MLP?]**
>
> Yes, the concatenation is fed into a one-layer MLP, which outputs the plausibility score.
>
> We have updated the paper to state that the concatenation is fed into an MLP classifier F^{cls}, as well as that the text embedding is computed using a Transformer text encoder F^{text}. Please refer to *Section 2: Problem Setting -- Paragraph "Commonsense QA"*.
>
>
> **2.3: In both graph encoders, the subgraph G(q,a) together with the aggregation weights - [neither of the descriptions mention aggregation weights]**
>
> To improve the presentation here, we have incorporated the following changes into the updated paper:
> - In the Commonsense QA part of Section 2, we removed the description of GNNs and only consider path-based models (which include RN and MHGRN), since path-based models are the ones designed for interpretability.
> - We compressed our description of path-based models to be self-contained and convey only the essential message: *“...path-based models compute the graph embedding by using attention to selectively aggregate paths in the subgraph. The attention scores can help explain which paths the model focused on most for a given prediction.”*
>
> Given our paper’s focus, we felt that detailed explanation of path-based models would not be very beneficial. Also, in the original submission, by “aggregation weights”, we were referring to the attention scores used for path aggregation.
>
> Please refer to *Section 2: Problem Setting -- Paragraph "Commonsense QA"*.
>
>
> **2.4: while the task is to predict the unobserved interaction (yuv = 0). - [does the zero here mean that the interaction is unobserved, or that the user has not engaged with the item? (or both?)]**
>
> y_{uv} = 0 means that the interaction between user u and item v has not been observed. In this case, user u may or may not have engaged with item v in the past. For (u, v) pairs where y_{uv} = 0, our goal is to predict how likely user u is to want to engage with item v.
>
> We have clarified this in the updated paper. Please refer to *Section 2: Problem Setting -- Paragraph "Item Recommendation"*.

---

> > ### Author Response · Authors · 2020-11-19
> > **Response to AnonReviewer2 (Part 2)**
> >
> > **2.5: where the aggregation weights are personalized for u - [you need to define how the aggregation works]**
> >
> > For reference, we include basic descriptions of KGCN and RippleNet here (but not in the paper):
> > - In KGCN, we first retrieve u and v from G’s learned embedding table. Next, a neural network updates v by iteratively aggregating v’s subgraph of L-hop neighbor items. At each hop h ∈ [1,...,L], an attention mechanism scores every h-hop neighbor with respect to both u and the neighbor’s relation to v, then aggregates neighbors into an updated v based on these attention scores.  After v is done updating, we compute <u, v>.
> > - In RippleNet, we retrieve from G’s learned embedding table, then set the items u has interacted with as seed items.  Next, a neural network computes u by using attention to iteratively aggregate the seed items and their L-hop neighbors, with respect to the relations between item pairs. After obtaining u, we compute <u, v>.
> > - In both KGCN and RippleNet, the attention scores indicate which neighbor items/relations the model judges as most relevant to the given user-item pair. As in commonsense QA, high-scoring neighbor items/relations can be used to explain the model’s prediction.
> >
> > To answer your comment, how aggregation weights are personalized w.r.t. u is addressed by this part of the KGCN explanation:
> > "At each hop h ∈ [1,...,L], an attention mechanism scores every h-hop neighbor with respect to both u and the neighbor’s relation to v, then aggregates neighbors into an updated v based on these attention scores."
> >
> > In the updated paper, we have compressed our description of KG-augmented recommender systems to be self-contained and convey only the essential message:
> >
> > - *“KG-augmented recommender systems differ mainly in how they use G to compute u and v. Generally, these models do so by using attention to selectively aggregate items/relations in G.  The attention scores can help explain which items/relations the model found most relevant for a given prediction.”*
> >
> > Given our paper’s focus, we felt that complete explanations of KGCN and RippleNet would not be very beneficial. Furthermore, KGCN and RippleNet are relatively involved (and not that similar to well-known paradigms like GCN). This makes giving proper explanations of both models kind of difficult, especially since we have limited space in the paper.
> >
> > Please refer to *Section 2: Problem Setting -- Paragraph "Item Recommendation"*.
> >
> >
> > **2.6: We randomly choose two triples (edges) in the KG and swap their relations. - [it would be good to see some actual examples here from one of the applications/KGs.]**
> >
> > In *Figure 2* of the updated paper, we show real examples of how each perturbation method changes the KG subgraph.
> >
> >
> > **2.7: where sG(·) is a KG scoring function trained on G. - [is this defined anywhere?]**
> >
> > The choice of s_{G} is task-specific, since KGs from different tasks may differ greatly with respect to semantics or connectivity. For commonsense QA, we use the scoring function from “Commonsense knowledge base completion” by Li et al. (2016). For item recommendation, we use the DistMult scoring function from Yang et al. (2015).
> >
> > We have added this explanation of s_{G} to the updated paper. Please refer to *Section 2: KG Similarity Metrics -- Paragraph "Aggregated Triple Score (ATS)"*.
> >
> >
> > **2.8: The font in table 1 is really small. If you get more room in a final version please turn this into two tables.**
> >
> > Following your suggestion, in the updated paper, we have split the commonsense QA table and the item recommendation table each into two tables (one per dataset).
> >
> >
> > **2.9: in-house test accuracy - what's an "in-house" accuracy?**
> >
> > Since the labels for the official CSQA test set are not available, we use the “in-house” test split introduced in “Kagnet: Knowledge-aware graph networks for commonsense reasoning” by Lin et al. (2019). This in-house test split has become a standard evaluation protocol for commonsense QA, used in prior works like “Scalable Multi-Hop Relational Reasoning for Knowledge-Aware Question Answering” by Feng et al. (2020) and “Connecting the Dots: A Knowledgeable Path Generator for Commonsense Question Answering” by Wang et al. (2020).
> >
> > We have clarified this in the updated paper. Please refer to the caption of *Table 1*.

---

> > ### Author Response · Authors · 2020-11-19
> > **Response to AnonReviewer2 (Part 3)**
> >
> > **2.10: We then compute the relation specific clustering coefficient vector c^r - do you define this anywhere?**
> >
> > Here is a more complete definition of c^r, which we've included in the updated paper:
> > - The SC2D metric is derived from the local clustering coefficient. For a given entity in G (treated here as undirected), the local clustering coefficient is the fraction of possible triangles through the entity that exist (i.e., how tightly the entity’s neighbors are clustering around the entity).  For entity e_i ∈ E, the local clustering coefficient is defined as c_i = 2Tri(e_i) / (deg(e_i) (deg(e_i)−1)), where Tri(e_i) is the number of triangles through e_i, and deg(e_i) is the degree of e_i. For each relation r ∈ R, let G^r be the subgraph of G consisting of all edges in T with r.  That is, G^r = (E, r, T'), where T′ = {(e, r, e′) | e, e′ ∈ E}.  Let c^r denote the |E|-dimensional clustering coefficient vector for G^r, where the i-th element of c^r is c_i.
> >
> > We have updated the paper so that the definitions of SC2D and SD2 are self-contained. Please refer to *Section 3: KG Similarity Metrics -- Paragraph "Similarity in Clustering Coefficient Distribution (SC2D)"*.
> >
> > **2.11: OpenBookQA (OBQA) - do you give a reference?**
> >
> > Good catch. We have added the reference to OBQA in the updated paper. Please refer to *Section 5.1: "Commonsense QA"*.
> >
> >
> > **2.12: We randomly select 10 questions - this seems like a small sample on which to perform the human evaluation. This evaluation would be more persuasive with a larger sample (even 20 or 30).**
> >
> > We agree that the human evaluation would be more convincing with a larger sample size.
> >
> > Following your suggestion, we have updated our paper to include 30 questions. Please refer to *Section 5.3: Auxiliary Experiments and Analysis -- Paragraph "Human Evaluation of KG Explanations"*.

---

### Author Response · Authors · 2020-11-19
**Summary of Changes**

We would like to thank all of the reviewers for their valuable comments. Based on the feedback we received, we have updated the paper in the following ways:
- **Major writing changes** in all sections to improve presentation clarity, such as cleaning up math, making explanations self-contained, removing extraneous info, and correcting typos. One of the biggest improvements was in the explanation of the DQN architecture and training procedure in Section 4.2.1.
- **Reorganized paper sections**: (1) merged the Pilot Study section into the Methods for Targeted KG Perturbation section; (2) put descriptions of KG similarity metrics into a new KG Similarity Metrics section; (3) moved auxiliary experiments into a new Auxiliary Experiments and Analysis subsection.
- **Added new figures** to better explain our method: (1) Fig. 1 depicts our high-level procedure; (2) Fig. 2 illustrates how a real ConceptNet subgraph changes in response to each of our perturbation methods; (3) Fig. 3 displays our DQN architecture for RL-RR.
-**Provided more substantial analysis**: (1) discussed interesting performance differences across different model/dataset/perturbation configurations (e.g., “Human Evaluation of KG Explanations” paragraph of Section 5.3); (2) explained counterintuitive perturbation results (e.g., “Why do perturbed KGs…” paragraph of Section 5.3).
-**Updated human evaluation** (“Human Evaluation of KG Explanations” paragraph of Section 5.3): (1) increased sample size from 10 questions to 30 questions; (2) provided real examples of KG explanation paths and their ratings; (3) changed rating dimensions from [Interpretability, Validity] to [Readability, Usability].
-**Clarified the big picture of our work**: (1) explicitly described our goal, problem setting, and high-level procedure in Section 2; (2) discussed motivation and implications in Sections 1 and 7.
-**Improved table readability** by splitting multi-dataset tables (with small font) into single-dataset tables (with large font).

We are happy to discuss any additional questions or concerns.

---

### Decision · Program_Chairs · 2021-01-07
**Final Decision**

**Decision:**

Accept (Poster)

**Comment:**

The paper's main message is that some existing NLP techniques that claim to improve performance by the use of a knowledge graph may not achieve this improved performance because of the knowledge graph or at least the explanation given may be questionable.  This is thought provoking and it will incite the community to think more carefully about the real factors of improved performance.  The initial version of the paper was not well written, but the authors improved the writing significantly.  The paper includes a thorough empirical evaluation to support the main message.  I have read the paper and I believe that this work will be of interest to a diverse audience.